# Transmission of 3D Holographic Information via Conventional Communication Channels and the Possibility of Multiplexing in the Implementation of 3D Hyperspectral Images

**Sergey A. Shoydin * and Artem L. Pazoev**

Department of Photonics and Device Engineering, Siberian State University of Geosystems and Technologies, 10 Plakhotnogo St., 630108 Novosibirsk, Russia; pazoev-al2018@sgugit.ru
* Correspondence: shoydin@ssga.ru

**Abstract:** This paper shows the possibility of transmitting 3D holographic information in real time with a TV frame rate over conventional radio channels by transmitting two two-dimensional signals in two image modes: depth map and surface texture of the object (mask + texture). The authors point out that it is similar to compression through eliminating the carrier and it is inherently similar to SSB (single-sideband modulation) but has higher resolution ability in reconstructing 3D images. It is also shown that such technology for transmitting 3D holographic information is in good agreement with the tasks of both aggregating and multiplexing 3D images when they are transferred from one part of the electromagnetic spectrum of radiation to another and the creation of hyperspectral 3D images.

**Keywords:** holography; digital holography; interference; holographic interference fringes; 3D photography; single sideband modulation; 3D television; 3D augmented reality



## 1. Introduction

The actual information capacity of holograms is so high that, in reaching for the goals set for creating holographic television, a number of stumbling blocks have been encountered, such as transmitting holographic information via available communication channels. Direct assessment of the amount of information when transmitting 3D signals frame by frame shows that one such holographic channel covers the whole radio frequency region, including all radio and TV channels. Only fiber optic channels of telecommunication and direct transmission systems with laser beams may theoretically be capable of transmitting such large volumes of information that dynamically vary from frame to frame. However, first, such broadband waveguide channels are limited by both their hardwired network or direct view distance, and second, by the fact that they are still in the pre-design stage.

This paper shows the possibility of transmitting full-fledged 3D holographic information via conventional communication channels by transmitting two two-dimensional signals, representing the 2D texture of the surface of a 3D object and a 2D map of its surface (mask). The introduction (Section 1.1) discusses the need to find ways to compress highly informative holographic information and presents an overview of modern works close to this topic (Section 1.2) and a methodology for describing holographic processes in the language of spatial frequencies (Section 1.3). The results of numerical analysis of spectral texture and mask properties are demonstrated (Section 2.1) using a case of a real 3D human portrait image. A hologram-creating technique is also described (Section 2.2). A hologram case is used to explain the reason for inefficiency of direct mathematical coding of hologram structure and the problem of observing parallax under the frame of the applied mathematical model using the spatial frequencies language (Section 2.3). The advantages of transmitting holographic information about a 3D object with two 2D signals are also highlighted (Section 2.4). Section 2.5 presents the results of direct transmission of such signals over a conventional communication channel with a frame frequency of TV, which



ensures the creation of a hologram at the receiving end of the channel and the restoration of a 3D image from it. In the last section (Section 2.6), some applications are considered simply, without unnecessary mathematical transformations, arising from the technology under consideration, including in the field of hyperspectral images and multiplexing of 3D images. Finally, in Section 3, the main results of the work are formulated.

### 1.1. Density of Recorded Holographic Information

Practically right after the first manuscripts on holography were published [1], researchers started focusing on the potential such holograms have for storing information. Research studies [2,3] described the first optical circuits in dedicated holographic storage devices based on the fact that a hologram writes down practically all of the data on the monochromatic wavefront that is used for recording information. An image restored with a hologram has transverse resolution $h$, as in any traditional system of image transfer, which is defined with the so-called Rayleigh criterion [4], equal to the Airy disk diameter Equation (1):

$$h \sim \frac{\lambda^2}{D/f}, \tag{1}$$

where $\lambda$ is the emission wavelength, $D$ is the transverse size of the hologram, and $f$ is the distance to a point of the restored image. Owing to the high resolution, the order of magnitude of the amount of data $N$ that can be written to such a hologram [2,3] can be assessed with Equation (2). Further practical studies [5–13] made significant allowances which were related, on the one hand, to the constraints on data density in a holographic storage device and, on the other hand, to finding ways to increase recording density. Potential data recording capacity $n$ in a hologram can be estimated approximately according to Equation (2), based on [5–10], from which it can be defined as the number of the points the size of which is defined by the Airy disk diameter Equation (2):

$$n \approx \Omega^2 \left(\frac{D}{\lambda}\right)^2 = S^2 \left(\frac{\Omega}{\lambda}\right)^2, \tag{2}$$

where $\Omega = D/f$ is relative aperture of the imaging system, in this case a hologram, and $S$ is its area. It follows that when $\Omega = 1$, which is typical of a good optical imaging system, the data recording density in a hologram corresponds to approximately one bit over the area equal to $\lambda^2$, which in the visible range corresponds to a spatial frequency of two to three lines per μm.

For pictorial holography of A4 sized (210 × 297 mm) portraits of real people, which can be compared to a 14″ TV screen, when $\lambda = 0.63$ μm, the number of points in a hologram reaches $n = 1.6 \times 10^{11}$, which in binary recording is equal to the same amount of data measured in bits. That is approximately $n = 2 \times 10^{10}$ bytes $\approx 20$ GB and is equivalent to the amount of information contained in a 60 min Full HD movie. Transmission of such arrays with a refresh rate of 25 Hz needs channels with throughout capacity $C = 5 \times 10^{11}$ B/s.

The bandwidth in this case is more than 100 GHz, which would more than cover the radio band available now. Actually, some of the first holographic TV developers [14] wrote: "Due to high specific density of holographic information, it is necessary to register and transmit a great number of discrete elements of holograms via communication channels, and their resolution and processing rate exceed the operational capabilities of existing TV devices and standard telecommunication channels." In this regard, Yu. N. Denisyuk, one of the founders of holography [15], pointed out that we do not yet know the fundamental principles of holography required for creating new types of 3D films and artificial intelligence.

Large volumes of holographic information, which are undoubtedly a great advantage in other spheres, in the domain of holographic TV and augmented reality play a dirty trick on developers, and this has come to be one of the critical deterrents to transmitting holographic information via telecommunication channels. The solution to this challenge

would be to either reduce the size of the holographic image significantly (which is hardly compatible with the objectives of augmented reality) or compress the holographic data in a way that the volume would not exceed reasonable limits (i.e., the capability of state-of-the-art 3G and 4G communication channels). The paper focuses on solving this problem.

### 1.2. Overview of Modern Works

The complexity of the task postponed the implementation of a full-fledged 3D holographic TV and a number of 3D augmented reality tasks. Stereo image broadcasting systems that do not require a large volume of transmitted information occupy only two times the frequency band of the communication channel. However, they were rejected. Relatively recently (in 2016), the TV companies Samsung and LG [16] phased out stereo TVs. One reason is the inconvenience of viewers having to use glasses. Sometimes, helmet-mounted TV stereo attachments are mistakenly called holographic monitors [17]; they are also inconvenient to use.

Another relatively economical method of transmitting volume to TV is Pleno technology, well developed by the Joint Photographic Experts Group (JPEG) [17]. In a traditional form, such as with the use of a microlens plate, it is poorly suited for presenting a classic 3D TV signal, since it does not provide reproduction of a large image volume. It has proven itself only as a method of adjusting the focus for non-sharp photo and video shooting.

The transmission of information of 3D objects using their representation in holographic form, when a dynamic 3D image (at least 25 frames per second) is formed directly in the space in front of the monitor, where the observer is located, could be the start of a new era of television and augmented reality.

The decision to present holographic content by the Joint Photographic Experts Group has not yet been made, although many options are being considered, and open databases of holographic content have been created to test methods proposed by various researchers for testing 3D images, compressing the information stored in them and reproducing them, preferably without loss, or at least with low loss, as in classic JPEG encoding. All of these examples of open holographic content have low spatial resolution of the reconstructed image and are difficult during digital processing with the 25 frames per second frame rate required for TV and 3D augmented reality. However, these examples do not provide a method for obtaining them. Additionally, it can be reasonably assumed that none were transmitted over a conventional radio channel with a frequency of at least 25 frames per second.

The same group of respected experts has planned work toward the creation of JPEG Pleno, developing a standard that will facilitate the capture and representation of light fields in model of point cloud, similarly to the research in this direction [18] and the works in holographic image modes [17]. Work is actively underway toward saving computer resources when compressing holographic images in JPEG format using deep learning [19] and other methods of saving computer resources when calculating optical transformations [20].

However, the problem of transmitting a holographic signal over a conventional communication channel was not directly discussed. Basically, all researchers agree on one thing: for the current level of computing technology, its performance is still insufficient for processing, and, as can be understood, for transmitting large amounts of holographic data [21–23].

The technology for 3D object hologram synthesis closest to that described in this paper is in [24], in which a hologram is also formed by a pair of frames with mask + texture, although in a different way, by convolution with the impulse response of the system. This method consists of two stages: obtaining an impulse response and then convolving with it. We will use another method based on the Fresnel transform, which is carried out in one stage of calculations. In addition, questions about the transmission of holographic information through communication channels, the properties of multiplexing, and hyperspectral capabilities are not addressed in [24].

It is noteworthy that, today, the main efforts of developers are focused on mathematical methods of processing received holographic signals, often divorced from physical processes. This consists of polygonal and voxel graphics, born from the idea of a point cloud, only in the first case an elementary object from a point is modified into a triangle, a network of which covers the indivisible surface of a 3D object [25–27], and in the second into cubes, an array of which creates a 3D picture of a three-dimensional, dynamically changing object, with the possibility of expanding it, such as the creation of splashes from a moving and developing wave [28–30].

Such technologies require significant computing resources. For example, the complexity of $N$-point cloud technology increases proportionally to $N^3$, polygonal to $N^9$, and voxel even more, which is a serious obstacle to their practical implementation. They are mathematical models of real space, far from the physical essence of transmitted signals.

In our opinion, relatively little attention has been paid to the physical process of registering 3D holographic information more precisely to the object-oriented computational process of creating holograms. The rapidly increasing speed of processing any information has created the illusion that soon polygonal arrays and even voxel arrays will be processed as easily as point clouds (although the latter is not yet suitable for holographic TV in terms of processing speed). However, the actual transmission of such arrays over a distance requires new communication lines, such as an optical cable, over which the spectrum of the transmitted signal exceeds the radio range by orders of magnitude. Direct transmission of a full-fledged 3D holographic signal, a point cloud, or a polygonal and voxel signal via conventional radio communication channels is impossible in principle, since one such transmission occupies the entire radio range available to mankind.

In this paper, a more physical approach is considered, which is a method of photographically registering the surface texture of a 3D object and a map of its surface depths, which comprise the initial information for generating a full-fledged hologram at the receiving end of the communication channel. This method is somewhat similar to the point cloud method, but it is simple and computationally much less laborious. It is suitable for dynamic reproduction of holograms reconstructed at the receiving end of the communication channel, and the field of view is quite large.

Most importantly, it allows us to transmit the necessary 3D holographic information over a conventional radio channel and create a 3D hologram after receiving the signal at the receiving end of the communication channel. Informationally, it is similar to the well-known SSB (single-sideband modulation) method, but it has some advantages (for example, in resolution). An analysis of the comparative spatial-frequency characteristics of the 3D image transmission methods of texture + mask and SSB speaks in favor of the first method.

Using holographic images when creating virtual 3D reality is the most promising, since they are basically adequate for the physical structure of the light field of the holographic object. Close to this is the method of transmitting 3D information considered in this paper using two two-dimensional pictures: the surface texture of the 3D object of holography and a depth map of the surface (mask) of the transmitted 3D image. In this method, a hologram is created already at the receiving end of the communication channel, which significantly reduces the amount of information transmitted. Such a hologram is a full-fledged analogue of the classical one and can restore not only imaginary 3D images observed behind the monitor screen but also real ones observed in front of the screen, directly in the volume of space where the observer is located. It is this advantage that allows us to create 3D virtual reality. A related effect is the ability to record and economically transmit hyperspectral images over the communication channel, expanding the range of the visible spectrum.

This paper is devoted to the study of the features of one method of representing 3D signals and transmitting them over a conventional, non-broadband communication channel, but allowing the synthesis of a full-fledged hologram at the receiving end, restoring a 3D image of a holographic object with the high resolution of a Full HD or even 4K frame.

### 1.3. Amplitude and Phase Modulation in Holography

To discuss the options for transmitting 3D holographic information, it is convenient to use the terminology of the spatial frequency spectra of holograms. Since the material realization of synthesized digital volumetric reflective holograms according to the scheme of Yu. N. Denisyuk [1] with the current state of technology is not possible, we will consider, without limiting the generality, the classical scheme for recording transmitting analogue holograms according the scheme of E. N. Leith and J. Upatnieks [31], presented in Figure 1. In terms of the linear response of holographic material, the structure of interference fringes is formed with firing object and reference beams Equation (3):

$$I(x_1, y_1) = |U_1(x_1, y_1) + R(x_1, y_1)|^2,$$
(3)

where $U_1(x_1, y_1)$ and $R_1(x_1, y_1)$ are complex amplitudes of the electromagnetic light field in the hologram recording plane $(x_1, y_1)$, which in scalar approximation of diffraction theory represent object and reference beams, respectively. When recording such a fringe pattern onto high-resolution photosensitive material under the effect of $I(x_1, y_1)$, photoresponse $\Delta\Psi(x_1, y_1)$ takes place, which is variable in the hologram field. In terms of the amplitude response of the holographic material, it is a material transmission variance $\tau(x_1, y_1)$, while for a phase response it is an alteration in eikonal $\Delta\varphi(x_1, y_1)$, which occurs due to either a change of local thickness in the hologram field $\Delta l(x_1, y_1)$ or the refraction index varying over the hologram field $\Delta n(x_1, y_1)$, or combination of all three mechanisms in the ratio defined by the physical and chemical properties of the holographic material itself [32].

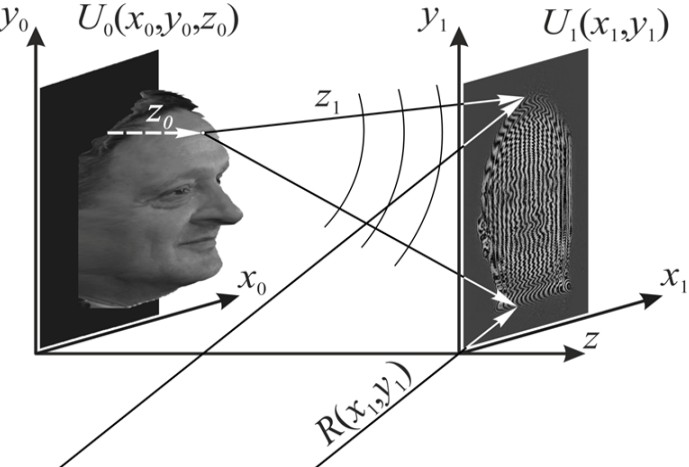

**Figure 1.** Hologram recording arrangement of E. N. Leith and J. Upatnieks. The complex amplitude of radiation scattered by an object forms object wave $U_0(x_0, y_0, z_0)$, which, propagating along z, forms complex amplitude $U_1(x_1, y_1)$ in the $(x_1, y_1)$ plane. As a result of its interference with reference wave $R_1(x_1, y_1)$ in the $(x_1, y_1)$ plane, holographic interference grating $I(x_1, y_1)$ is created, which forms a photoresponse $\tau(x_1, y_1)$ either in the holographic material or on the photomatrix. $z_0$ is distance from the plane where a 3D object is located to a point on its surface.

Further, for clarity and to avoid limiting the general nature of the basic conclusions in this paper, we take $I(x_1, y_1) << I_0$ over the whole hologram field, with a photoresponse $\Delta\Psi(x_1, y_1)$ as an amplitude coefficient of transmission $\tau(x_1, y_1)$.

The structure of such a hologram is written as Equation (4), where the multiplier prior to cosine constitutes the visibility of interference fringe $V$; $I_a$ and $I_r$ represent the intensity of the object and reference beams, respectively; and under the cosine there is local phase difference of $\varphi_a$ (object) and $\varphi_r$ (reference beams).

$$\tau(x_1, y_1) = |U_1(x_1, y_1) + R(x_1, y_1)|^2 =$$
$$= \{I_a(x_1, y_1) + I_r(x_1, y_1)\} \cdot \left\{1 + 2 \cdot \frac{\sqrt{I_a(x_1, y_1) \cdot I_r(x_1, y_1)}}{I_a(x_1, y_1) + I_r(x_1, y_1)} \cdot \cos[\varphi_a(x_1, y_1) - \varphi_r(x_1, y_1)]\right\}$$
(4)

Actually, the two phases change in space. In the reference wave, this often occurs just in linear fashion, when it is flat and falls at an angle to a normal line. In the object wave, it is formed with a Fresnel image from the signal formed by an object. The spatial spectrum of such a hologram is formed on the basis of the fundamental wave, which has the meaning of carrier spatial frequency $\omega = k \cdot \sin(\theta_r)$ equal to the average rate of change of phase difference in the space Equation (4), where $\theta_r$ is the reference beam angle to the optical axis. Deviation caused by the shape of a holographic object also has an impact on the spatial frequency of a holographic grating. The modulation depth of the spatial spectrum is defined by the visibility of fringe pattern $V$. The carrier frequency Equation (5) defines the rotation angle of a reconstruction beam tilt when it comes to a hologram:

$$\omega_{x_1} = \frac{\partial \varphi_r(x_1, y_1)}{\partial x_1} = k \cdot \sin(\theta_{x_1}); \quad \omega_{y_1} = \frac{\partial \varphi_r(x_1, y_1)}{\partial y_1} = k \cdot \sin(\theta_{y_1}) \tag{5}$$

while its deviation $\Omega$ is the diffraction spectrum of waves which form the image of the object recorded in a hologram Equation (6):

$$\Omega_{x_1} = \frac{\partial \varphi_a(x_1, y_1)}{\partial x_1} = k \cdot \sin(\Delta\theta_{x_1}); \quad \Omega_{y_1} = \frac{\partial \varphi_a(x_1, y_1)}{\partial y_1} = k \cdot \sin(\Delta\theta_{y_1}). \tag{6}$$

The hologram forms spatial harmonics of carrier frequency $\omega$ with wave-number vector $k$ and lateral harmonics $\omega \pm \Omega$ with wave-number vectors $k \pm \Delta k$, similar to the phase-modulated signal known in radio engineering [33].

They generate diffraction of the reconstructed wave into angles $\pm \theta_{x1}, \pm \theta_{y1}$ with respective periods $d_{x1}$ and $d_{y1}$. Likewise, deviation of spatial frequencies of the object wave generates gratings with periods from $dx_1{}^{\min}$ to $dx_1{}^{\max}$ and from $dy_1{}^{\min}$ to $dy_1{}^{\max}$ Equation (7):

$$\begin{array}{ll} d_{x1} = \lambda / \sin(\theta_{x1}); & d_{y1} = \lambda / \sin(\theta_{y1}); \\ d_{x1}^{\max} = \lambda / \sin(\theta_{x1} - \Delta\theta_{x1}); & d_{y1}^{\max} = \lambda / \sin(\theta_{y1} - \Delta\theta_{y1}); \\ d_{x1}^{\min} = \lambda / \sin(\theta_{x1} + \Delta\theta_{x1}); & d_{y1}^{\min} = \lambda / \sin(\theta_{y1} + \Delta\theta_{y1}). \end{array} \tag{7}$$

Therefore, when structure $\tau(x_1, y_1)$ is illuminated with a reconstruction beam, the radiation restored with a hologram has the key spatial harmonic with period $d_{x1}$, which defines the rotation angle of the beam, as well as the two lateral harmonics with periods from $d_{x1}{}^{\min}$ to $d_{x1}{}^{\max}$, the spectrum of which forms the object image in the plus first and minus first orders. A tilt angle of reference wave $\theta x_1$ is chosen considering the need to separate beams of zero and minus first order in space to have them at distance $z$ from the hologram (angle $\theta y_1$ is often chosen to be zero for convenience), while the angles limited by $\Delta\theta x_1$ and $\Delta\theta y_1$ are defined by the spatial spectrum of the object wave. In a somewhat simplified one-dimensional case for flat uniformity in terms of the sectional area wave, the holographic signal intensity range Equation (4) is similar to the one-tone signal spectrum Equation (8):

$$U_1'(\omega) = \left[ \begin{array}{c} \delta(\omega) + \\ +\left(\frac{V}{2}\right) \cdot \delta(\omega - k \cdot \sin\theta_{x_1} \pm \Omega) + \\ +\left(\frac{V}{2}\right) \cdot \delta(\omega + k \cdot \sin\theta_{x_1} \mp \Omega) \end{array} \right] \tag{8}$$

This represents the zeroth harmonic, which restores beam $R_V$, on which it generates the zeroth order of refraction with $\omega = kx\sin(\theta_V)$, where $k = 2\pi/\lambda$, $\theta_V$ is the restoring beam tilt angle to an optic axis, and two more beams, plus and minus first orders, responsible for restoring the actual image on the spatial frequency one $\omega = kx[\sin(\theta_V) - \sin(\theta_{x1})]$ and a inverted image from the frequency one $\omega = kx[\sin(\theta_V) + \sin(\theta_{x1})]$.

It is obvious that the spectrum Equation (8) is similar to the range of frequency-modulated radio signals, where the desired signal with spatial frequency $\Omega$ widens the range of carrier frequency $\omega$ to the right and left along the spatial frequency axis. In

addition, Equation (4) also shows some amplitude modulation. It becomes apparent in modulating the interference fringes with visibility coefficient *V*, which spatially widens the area of the zeroth and plus/minus first order harmonics and depends on distribution of object and reference signal intensities over the hologram field. Therefore, although there are some differences, in general Equation (4) shows that there are amplitude and phase modulations in the spectrum of the recorded hologram similar to amplitude and phase modulation in radio engineering.

Actually, the carrier spatial frequency $\omega$ is high. It can be compared to a reciprocal wavelength of the record, and its period is equal to only a few micrometers, while the object spectrum $\pm\Omega$ is defined by its degree of complexity. Even in high resolution (e.g., Full HD standard), the greatest possible spatial frequency is about 2000 lines over a screen area, and with a $36''$ screen (~0.9 m), it would be about three lines per mm. The difference of almost three orders of magnitude compared with the estimations in Equation (2) enable us to rely on high compression of holographic information when encoding it in one sideband (SSB method) [34]. The conventional method of horizontal and vertical deflection for transferring frame data point by point includes transferring all spatial frequencies of a hologram. With regard to redundancy during direct encoding of a holographic image, when transferring spatial frequencies where all the frequencies are encoded, including those that do not have information about the object for forming a hologram located between $\omega = \Omega$ and $\omega = kx\sin(\theta_{x1}) + \Omega$ is not justified, though conventionally this method can be used to transfer holographic information [14,17,35]. The greater the difference between the carrier frequency $\omega = kx\sin(\theta_{x1})$ and its deviation $\Omega = kx\sin(\Delta\theta_{x1})$ caused by the diffraction-limited beam divergence of the object wave, the more effective the information compression will be.

Consequently, similar to [34], it is possible to transmit not the whole phase-modulated signal with the spectrum Equation (8) representing the whole hologram, but only the component that corresponds to the single sideband (SSB) with the spatial frequencies $\Omega = kx\sin(\Delta\theta_{x1})$, bearing some information about the texture of the object for forming a hologram and its surface map, which is referred to as a mask in this paper.

For this, it is enough to have a photograph of the texture and the surface map taken with the help of any method (mask). For simplicity, we did not use the cutting-edge ToF method. Although it may have bright prospects, we chose a simpler method of structured light [36]. Significantly reducing the volume of transferred information and keeping things comparatively simple, the structured light method allows the opportunity to record information about a living 3D object in real time, to transfer pairs of frames via a conventional radio channel, and restore a 3D image at the receiving end of the telecommunication channel by synthesizing a hologram according to the received information and digital input of a spatial frequency into it.

In order to restore an image at the receiving end of the transmitting channel, it is required to re-input the carrier frequency. Since there is no simple method for adding a carrier frequency to a two-dimensional holographic signal yet, it is suggested in the patent [37] to synthesize a hologram by a numerical method at the receiving end of the communication channel using the accepted mask and texture.

Although it is shown above that transferring 3D information about a holographic object in the form of texture + mask is similar to transferring a spatial spectrum, these two methods have significant differences, such as in the resolution of the reconstructed image, which is shown below in the computational simulation. The spatial-spectral representation is simply convenient for understanding the processes occurring during the recording of holographic information, its transmission over the communication channel, and the restoration of a 3D image at the receiving end of the communication channel.

## 2. Computational Simulation

*2.1. Replacing a Hologram with Two 2D Images with Subsequent Transmission for Synthesizing a Hologram at the Receiving End of the Communication Channel*

For a numerical experiment, a Full HD image of an object was used, with dimensions of 2000 lines by 2000 columns (Figure 2).

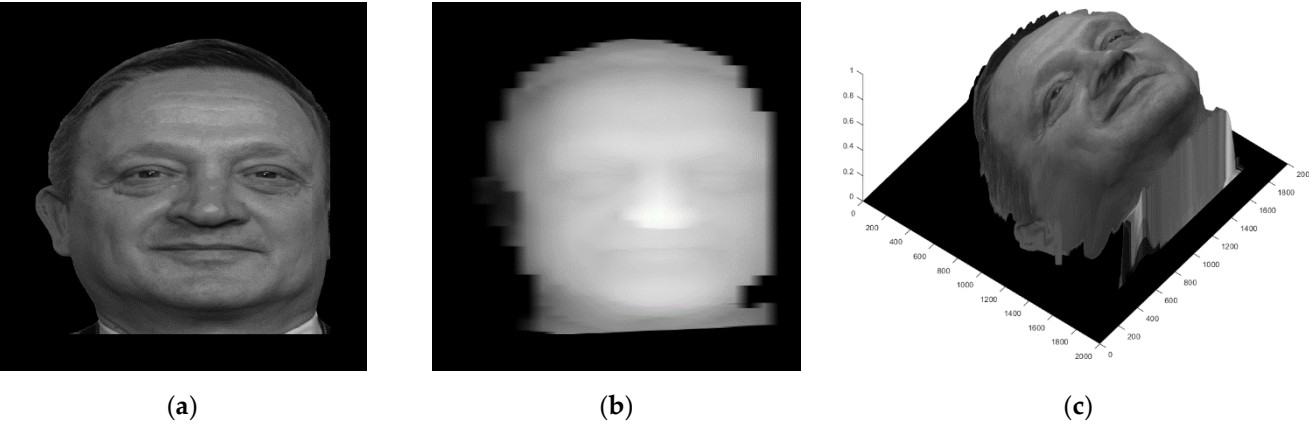

| (a) | (b) | (c) |

**Figure 2.** Structure of 3D holographic object: (**a**) surface texture; (**b**) mask; and (**c**) full 3D image of an object.

The stated color depth of the texture is 8 bits (256 tone gradations), which ensures quite adequate photographic quality of the 3D object surface. The technology described in [36] for creating a mask with the method of structured light, theoretically allows the use of a number of layers in the depth up to the size of half the discrete frame number, in our case up to 1000 layers. However, we would not need such high quality, so in further numeric experiments it was limited to 128 layers, according to the number of layers on which Fresnel transform would be applied when synthesizing a hologram. The Lloyd–Max quantizer can help reduce the number of samples in the mask depth or increase the entropy of samples required to encode it.

Figure 3 shows the spatial spectra of the object in Figure 2: its textures (a), masks (b) and the 3D object as a whole (c).

According to the figure, the mask spectrum (Figure 3b) at the level of 1/256 of the maximum, which can be considered as the noise level, is almost twice as narrow as the texture spectrum (Figure 3a), which indicates that the frame with a mask contains almost half the spatial frequencies in the horizontal plane and two to the second power (i.e., four times) less information.

In the case where a line of 2000 pixels can hold up to 1000 harmonics, it can be estimated that, with a signal/noise cut-off level of 1/256, the object mask in Figure 2b contains 62/2000 possible harmonics. That is about 3% of the information allowed in a Full HD frame (Figure 3b). The object texture in Figure 2a contains 116/2000, or about 6% of the possible information in a frame (Figure 3a). Even combined, they give a spectrum width less than 10% (Figure 3c). This proves that, in direct point-by-point (pixel-by-pixel) encoding, the biggest part of a signal does not carry any useful information. Even the high resolution of Full HD standard and even twice that in 4K are too much for the human eye to perceive such 2D images; nevertheless, it is not capable of transmitting the entire frequency band stored in a holographic image containing 3D object image.

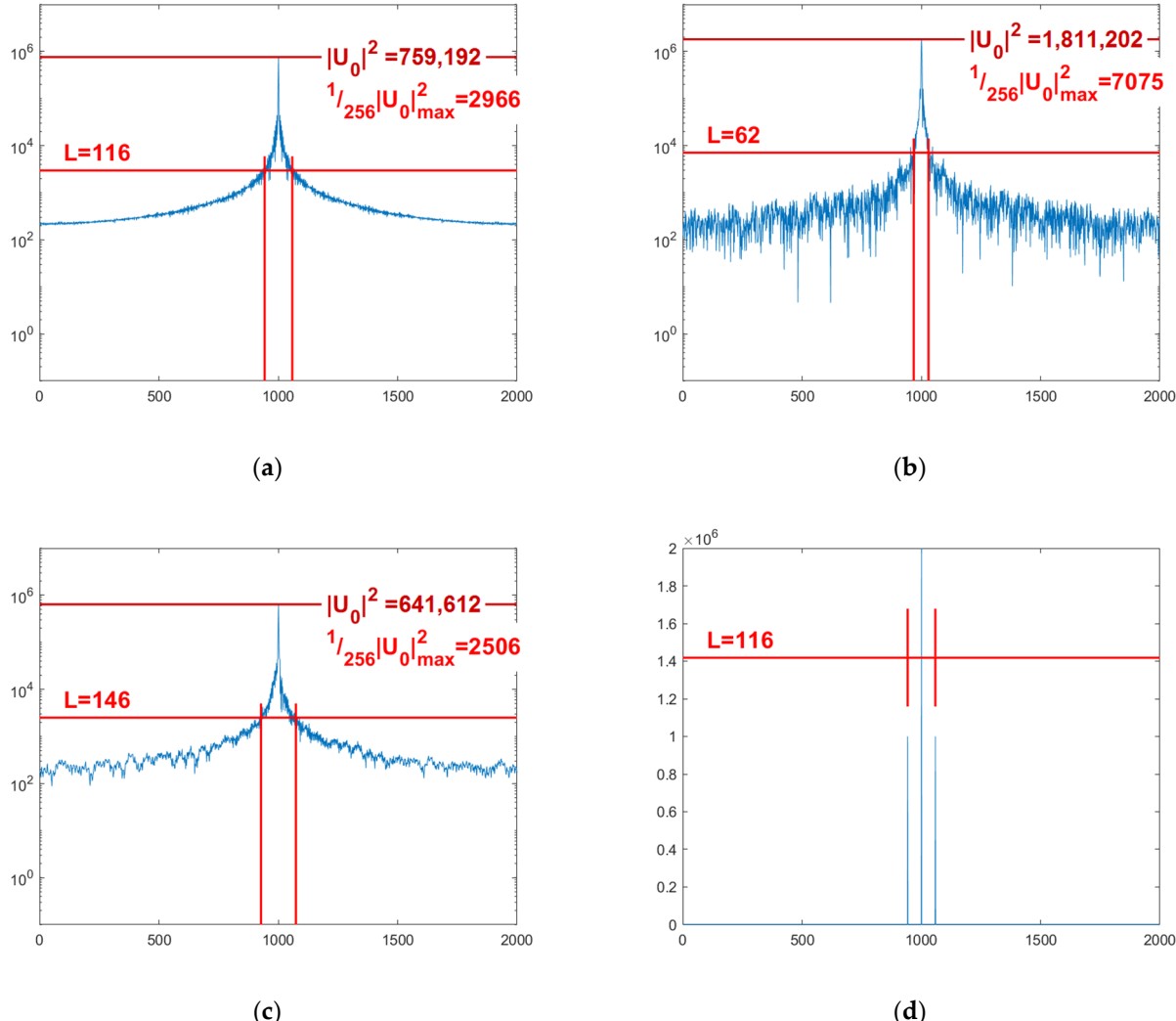

**Figure 3.** Horizontal section of spatial spectrum of images in Figure 2 for brightness depth level 1/256: (**a**) object texture; (**b**) object mask; (**c**) full 3D image of object; and (**d**) bearing sine wave with period equal to 34.5 pixels.

Judging by the spectrum, it can be seen that the very image of the surface of a 3D object is smoother than its texture (and in the frequency spectrum by at least two times). In reality, it may be more, since we usually do not notice the depth of wrinkles on a person's face, although we almost always see their projection on the surface texture. This also gives rise to an understanding of the inefficiency of direct 3D coding, proposed in one of the first patents for holographic TV [38].

### 2.2. Synthesizing a Hologram at the Receiving End of a Communication Channel Using Two 2D Images (Texture + Mask) and Computational Problems

Figure 4 shows a diagram of digital discrete synthesis of a hologram. By analogy with [39], we selected a method for splitting a 3D object into layers $U_{0m}$ (11), which was justified by the features of D-FFT (double fast Fourier transform), geared up to work with flat images. In order to fine-tune the software, a test model representing a half-tone gradual depth $z$ object $U_0$ was used. The process required getting a restored image of the minus first order of diffraction where the actual object was restored when diffracting. That is why a hologram field of a bigger area than that of the object was chosen to satisfy the requirements of the used Fresnel transform algorithm (D-FFT) [40], which requires using the same number of pixels in the hologram as in the object to form a hologram.

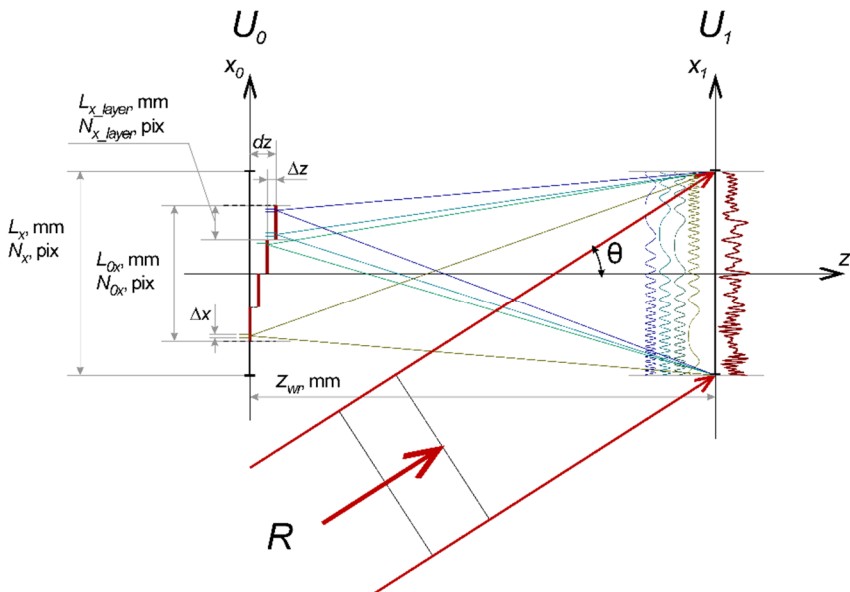

**Figure 4.** Diagram of digital discrete synthesis of a hologram. $U_0$, object; $U_1$, hologram plane; $R$, reference wave; $\theta$, reference wave tilt angle; $L_x$, object field width (mm); $N_x$, object field width (pixel); $L_{0x}$, object width (mm); $N_{0x}$, object width (pixel); $L_{x\_layer}$, object layer width (mm); $N_{x\_layer}$, object layer width (pixel); $\Delta x$, discrete width (pixel); $dz$, object depth; $\Delta z$, distance between $m$ layers; $z_{wr}$, distance between object and hologram.

According to the arrangement in Figure 4, a 2-stage algorithm for hologram synthesis was built. The first stage is a Fresnel transform Equation (9) for the amplitude of radiation diffused by a holographic object $U_0(x_0, y_0, z)$:

$$U_1(x_{1i}, y_{1i}, z_{wr}) = \sum_{x_{0i}, y_{0j}} \sum_{x_{1i}, y_{1j}} \sum_{z_{0m}} U_{0m}(x_{0i}, y_{0j}) \cdot \exp(ik(z - z_{0m})) \cdot \theta_{Fr}(x_{0i}, y_{0j}, z_{0m}, x_{1i}, y_{1j}) \quad (9)$$

where $k = 2\pi/\lambda$, $\theta_{Fr}$ is the kernel in the Fresnel transform Equation (10) for each of $m$ layers of object representation, and:

$$\theta_{Fr}(x_{0i}, y_{0j}, z_{0m}, x_{1i}, y_{1j}) = \exp\left\{\frac{ik}{2(z - z_{0m})} \cdot \left[(x_{1i} - x_{0i})^2 + (y_{1i} - y_{0i})^2\right]\right\} \quad (10)$$

where $z_{om} = z_{wr} + m\Delta z$ is the value of layer displacement $m$ along $z$. Here the 3D object for forming a hologram has the form $U_0(x_{0i}, y_{0j}, z_{0m}) = U_0(x_{0i}, y_{0j}) \cdot \exp\{ik[z - z_{0m}(x_{0i}, y_{0j})]\}$ and can be specified as the sum of layers of 3D object $\sum m U0(x_0, y_0, z_{0m})$, which, when using the fast Fresnel transform D-FFT$_{x1i, y1j}(x_{0i}, y_{oj})$ for each layer m at depth $z_{0m}$ [40], looks like Equation (11):

$$U_1(x_{1i}, y_{1i}, z_{wr}) = \sum_{z_{0m}} \left\{(D - FFT)_{x_{1i}, y_{1i} z_{0m}} \left[U_0(x_{0i}, y_{0j}, z_{0m})\right]\right\} \quad (11)$$

The second stage involves summing the obtained object beam with the reference beam and finding the transmission coefficient of the hologram $\tau(x_1, y_1)$ according to Equation (8). It should be pointed out that the tilt angle of the base beam should be selected to meet the Nyquist condition on the number of discrete samples per period of a holographic grating and the condition for separating a restored image (minus first order of diffraction) from a zero order diffraction beam that passed the hologram.

In our case, for a numerical experiment the following were chosen: $L_0$ equal to 10 mm, made up of $10^4$ dots, and $L_x$, for observing the parallax, equal to 47.5 mm, made up of 47,500 dots, and $\lambda = 0.532$ μm, further replaced with $\lambda = 0.633, 0.435$, and $0.937$ μm. When

the reference wave tilt angle was equal to 9°, $z_{rec}$ for such wavelengths was chosen to be about 80 mm.

In order to present the depth, we made a number of object layers according to their depth. The option of cutting a 3D object into 64 layers seemed to be not smooth enough. The Lloyd–Max quantizer helped to reduce the number of samples in the depth of the mask, thereby increasing the entropy of samples required for encoding it, but already 128 evenly spaced layers in our case were quite good and smooth enough along the depth image. So, we chose it to show that it is possible, in principle, to synthesize such holograms of living 3D objects, but not their maximum quality. Additionally, we could not use the direct Fresnel transform for a 3D object, as it requires higher computational power. Nevertheless, we did not challenge ourselves to make an authentic restored image of a hologram. We needed to prove the possibility of using a full recording cycle for a 3D image of a living person with Full HD quality and transmitting such information via a conventional telecommunication channel for restoration into a holographic 3D image.

Further improvement of the method requires the application of resources and the joint effort of various development teams, and, most of all, the development of economical algorithms for Fresnel transform of dynamic changes with the frame rate of 3D signals, which is now receiving more attention [41,42].

*2.3. Conditions for Forming a Hologram with the Necessary Parallax of the Restored Image and the Reason for the Permissible Compression of Holographic Information*

Figure 5 shows a hologram computed with the method shown above, Figure 6 shows its spectrum, and Figure 7 shows an image restored with it.

It is apparent that the hologram in Figure 5 is made up of interference fringes diffused with a Fresnel transformation formed in groups along an interference fringe by points of varied brightness. Interference fringes are present not only in the central part of the hologram, where the interference structure of the object distorted with the Fresnel transform can be seen, but also outside this area. The interference fringes outside the central section are in charge of image restoration, which is seen at an angle to the optical axis. For ease of parallax observation, a 1.5 zoom-in was made along the vertical ($y_1$) for the hologram field, and a 4.75 zoom-in along the horizontal ($x_1$). Therefore, the parallax (Figure 8) reached 4° vertically ($y_1$) and 29° horizontally ($x_1$). The image restored by the hologram "hangs" in the observer's space, in front of the hologram.

The spectrum of the obtained hologram ($\tau = |U_1|^2$) is shown in Figure 6. Figure 6a shows the two-dimensional spatial spectrum, in which it is seen that between the zeroth spatial frequency and the plus/minus first orders of diffraction, there is a significant interval, as was predicted with the estimates in Section 1.3. This distance in the frequency space points out that direct point-by-point encoding of holograms is not effective, as it also encodes empty areas of the space. The requirement for spatial separation of restored (minus first) and past (zeroth) orders of diffraction when restoring images, in this case, contradicts the requirement for data compression. To be more accurate, the requirement for separating beams is set for a hologram in our conventional three-dimensional space, while the requirement for data compression is for the reverse dimensionality space (i.e., frequency space).

These requirements are generated by different physical factors and conflict with each other. First, they stem from the requirement for comfortable image perception, which is expressed in a rotation angle measure of the minus first beam θ and sets the spatial carrier frequency. Second, they stem from the objective to fill in the frequency space with the object spectrum to the maximum degree possible. The first can be regulated in a schematic way, the second not, as it is defined by the composition of a holographed space with an object for forming a hologram, which we cannot manage with schematics. Similar to a radio signal, in order to eliminate excessive redundancy of the holographic signal, it is necessary to artificially separate the carrier and side spatial frequencies with the subsequent transmission of only one side frequency band to the receiving end of the communication

channel, which in our case is made in the form of transmitting a pair of 2D texture + mask images.

Actually, a conclusion can be made that although in conventional space it is impossible to bring the positions of images of the zeroth and first orders of diffraction closer, in the spectrum space the zeroth and plus/minus first orders can be placed much closer to each other. This means that it is theoretically possible to compact the information recording; the only thing is that an acceptable way must be found. For that, both the method of extracting one sideband minus the first order of the spatial spectrum of the object, as a kind of holographic analogue (SSB), and the method of transmitting two 2D images as texture + mask are suitable. However, for analysis, it is easier to use the well-known terminology of transferring information to SSB.

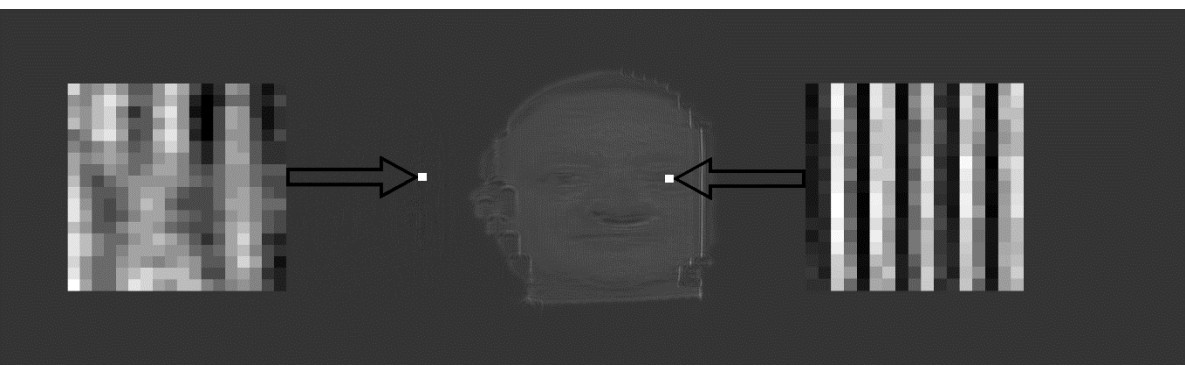

**Figure 5.** Hologram sized $15 \times 47.5$ mm with increment of 1 μm, $\lambda = 0.532$ μm, reference beam tilt angle of $9°$, synthesized at 74 mm from the object (zoomed-in fragments of hologram are shown).

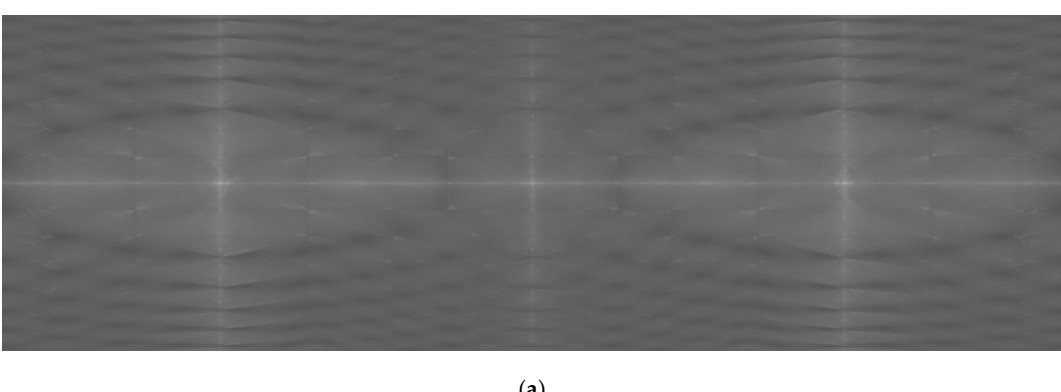

(**a**)

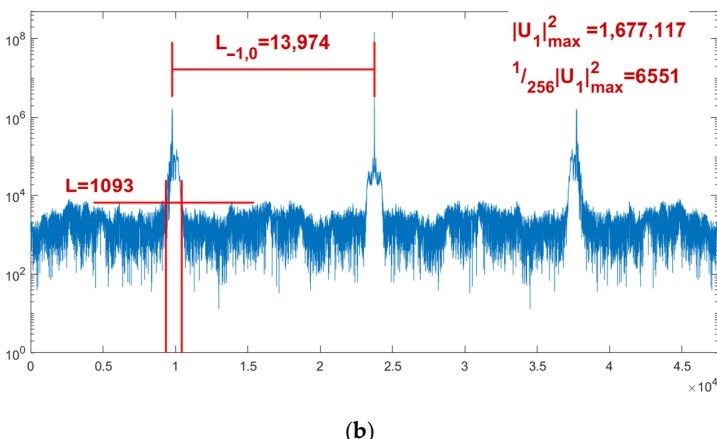

(**b**)

**Figure 6.** Hologram spectrum: (**a**) spatial arrangement and (**b**) logarithmic scale of spectrum intensity along horizontal

X-axis. Central peak of the spectrum is on the optical axis and corresponds to the 23,751th pixel from the hologram edge. The distance from the central peak of the zero order of diffraction to minus first order $L_{-1,0}$ = 13,974.

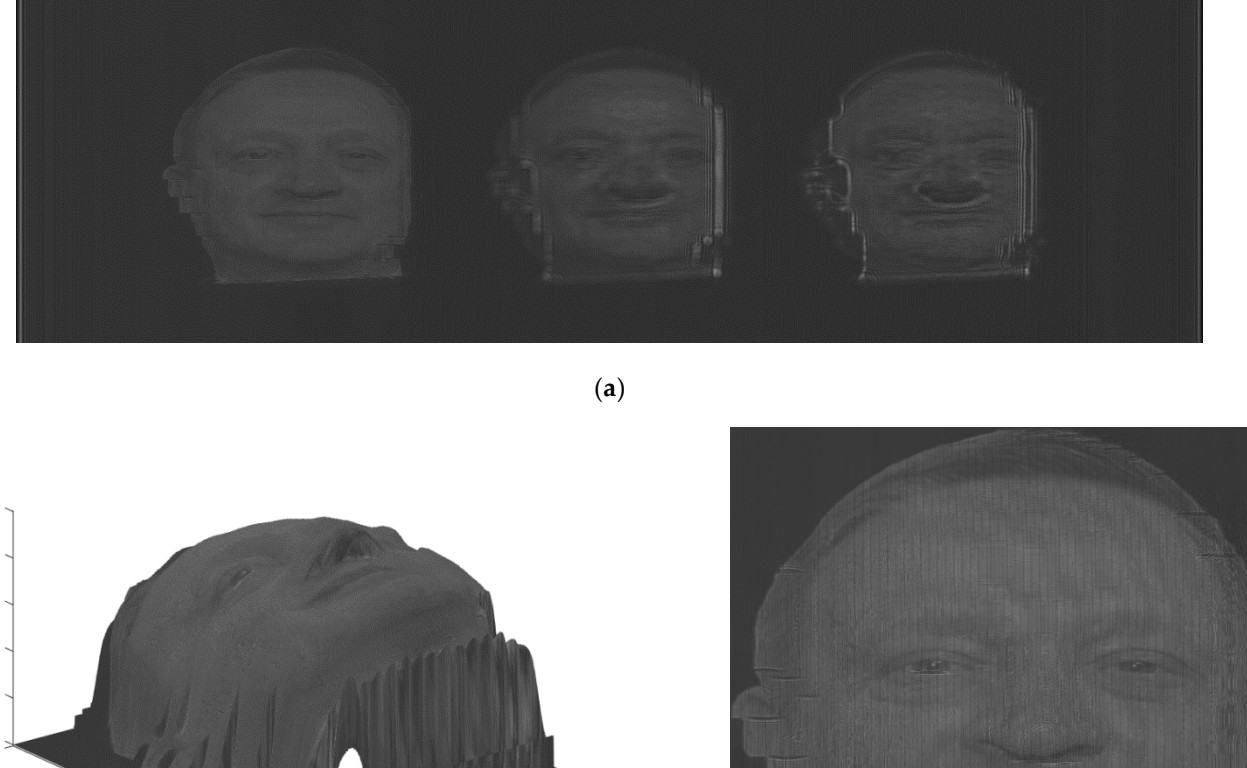

(**a**)

(**b**)

(**c**)

**Figure 7.** Volume of restored image. (**a**) Three orders of diffraction restored with hologram: minus first order (**left**), zeroth (**center**), plus first order (**right**). Restored 3D image made up of 128 layers: (**b**) isometric projection, (**c**) top view.

Figure 6a shows the spatial spectrum of the hologram in Figure 5, and Figure 6b shows a slice of this spectrum along the $x_1$ axis. It is apparent that at the same noise cut-off level (1/256) as in Figure 3c, the ratio of the spectrum width measured in pixels of the minus first order ($L$ = 1093) to the whole spectrum width (47,500) is equal to 2.3%. This means that the holographic signal can be compacted almost 43 times along the horizontal axis of the hologram, while along the vertical axis, the hologram dimension is 15,000 dots (15,000/1093 ≈ 14), which in total will allow us to compact it 43 × 14 ≈ 600 times. It also confirms that transmitting two main 3D image modes (texture + mask) rather than a hologram via the communications channel is a correct choice.

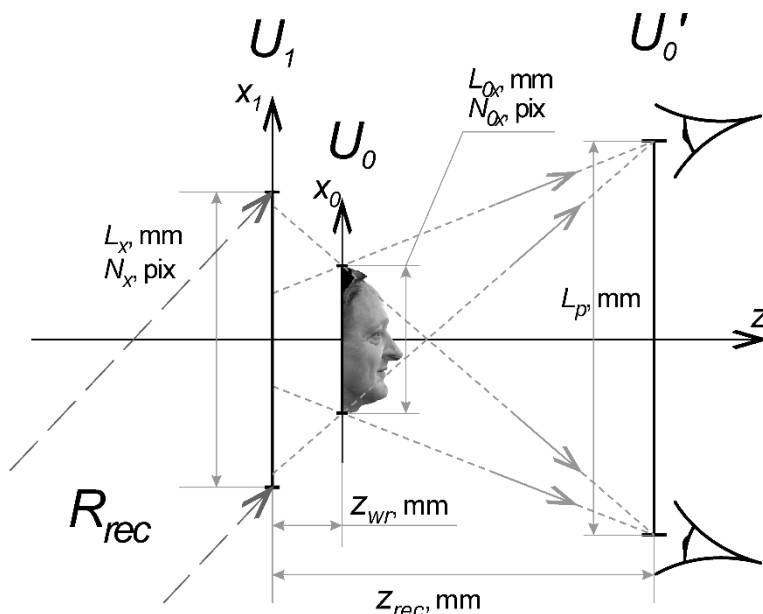

**Figure 8.** Conditions for observing parallax: $U_0$, object image; $U_1$, amplitude of wave restored with hologram; $U_0'$, amplitude of restored wave in observation plane; $R_{rec}$, restoring wave; $L_{0x}$, object width (mm); $L_x$, hologram width (mm); $N_{0x}$, number of object pixels in section x; $N_x$, number of hologram pixels in section x; $L_p$, parallax observation base; $z_{wr}$, distance between object and hologram; $z_{rec}$, distance between hologram and observation plane.

The spatial spectrum of the synthesized hologram, similar to its filled density, differs greatly from the restored image. It can be seen that there is a big interval left between the zeroth and plus/minus first orders of diffraction. This refers directly to the possibility of compressing holographic information theoretically predicted earlier in the paper. Again, we can conclude that in the spectrum space, the zeroth and plus/minus first orders can be placed much closer to each other (i.e., the information recording can be greatly compacted). Instead of allocating one side band of the minus first order of the object's spatial spectrum as a kind of holographic analog (SSB), we chose the method of transmitting two 2D images of the mask and the surface texture of the 3D object. This includes transmitting both the amplitude and phase components of the spatial modulation of the object wave to the receiving end of the communication channel. Later, at the receiving end of the communication channel, a hologram was synthesized using them, and by the following Fresnel transformation, the image of a 3D living object was restored.

Diffraction of the restoring beam in a hologram (Figure 7) forms (from left to right) a restored object image and the zeroth and plus first order of diffraction. The image restored by a hologram in Figure 7a is made up of 128 layers, and Figure 7b it is made up of 64 layers, which is why the farther layer is better seen on the left, while on the right the nearest layer of the 3D object is more visible.

Figure 7c shows an isometric projection of the restored 3D image of the object restored with the help of the hologram, which was recorded with the use of 128 layers. It is apparent that the volume of the hologram synthesized at the receiving end of the telecommunication channel was restored.

Further improvement of the depth transfer quality using the layer-by-layer application of the numerical Fresnel transformation algorithm (D-FFT) is possible by using the Lloyd–Max quantizer [43], which increases the entropy of samples and by unwrapping of the hologram phase, for example, by the Kalman method [44].

### 2.4. Advantages of Transmitting 3D Images by Pairs of 2D Images of Texture + Mask Compared with Transmission of the Spectrum of a 3D Object Using SSB

Much research has been devoted to comparing the quality of the transmitted image for JPEG compression of flat pictures, which cannot be said about 3D images, although the Joint Photographic Experts Group committee [17] plans for this work to be done. To compare the quality of the volumetric image reconstructed by the hologram, numerical experiments were carried out simulating two transmission algorithms.

The first is the transmission of the spectrum, when first a hologram is synthesized from a 3D object, then its two-dimensional spatial spectrum is calculated. From this spectrum of the hologram, a spatial-frequency region is distinguished, corresponding to the minus first order of diffraction. The image of this spatial frequency region of the spectrum is cropped to the frame size of the selected standard for image transmission, for example, JPEG, and transmitted over the communication channel. At the receiving end of the channel, a two-dimensional image of the spectrum is formed, and then the structure of the hologram is restored by inverse Fourier transform. At the last stage, the 3D image is restored with a hologram. It can be seen that in this case, the carrier spatial frequency is also eliminated.

The second way is to first transmit two two-dimensional signals (the above-described pair of images, texture + mask) via the communication channel. Then, at the receiving end, synthesize a hologram and restore a 3D image with its help. This way not only is simpler and shorter but also, as shown below, has higher resolution.

Figure 9 shows an image of a 3D test object, on each step of which an image of a linear raster with a variable step is plotted. The resulting measuring scale has 17 bands, decreasing in size from 256 pixels to 1 pixel.

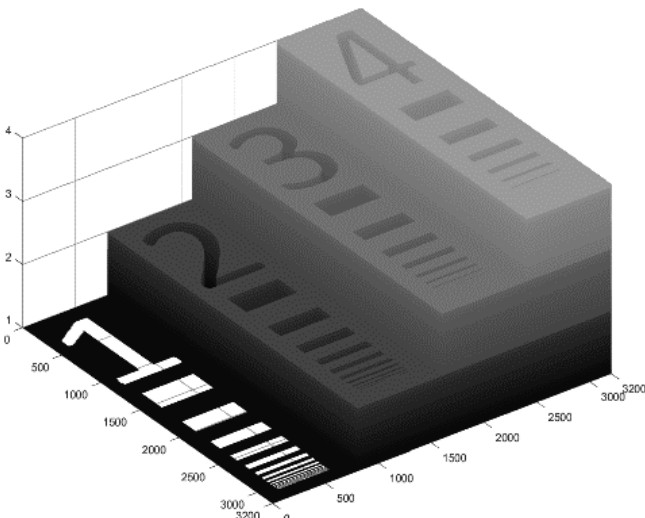

**Figure 9.** Four steps with a base of 6.4 × 6.4 mm, each 1.6 mm high, which in pixels corresponds to a base of 3200 × 3200 and a height of 800 pixels. Width of measuring scale strips in pixels is 256, 181, 91, 64, 45, 32, 23, 16, 11, 8, 6, 4, 3, 2, 1, and 1, corresponding to width in mm in our calculations of 0.512, 0.362, 0.182, 0.128, 0.090, 0.064, 0.046, 0.032, 0.022, 0.016, 0.012, 0.008, 0.006, 0.004, 0.002, and 0.002, respectively.

Figure 10 shows the results of calculations for measuring the scale in Figure 9. To illustrate the quality of the transmitted image, the second step from each restored image of the object is selected. The remaining three steps exactly repeat the restoration quality shown for step 2. The number of pixels is proportional to the selected area of the spectrum.

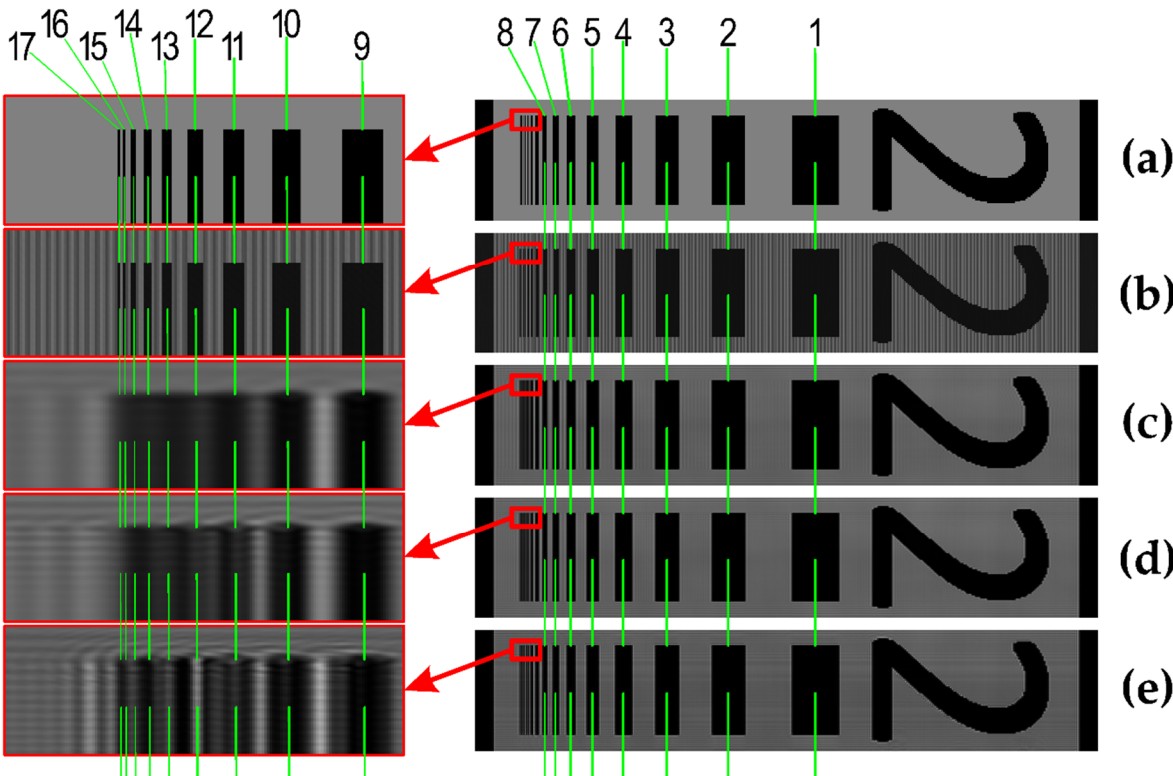

**Figure 10.** (**a**) Second step of test object (Figure 9); (**b**) second step of image of test object restored from hologram; (**c**) second step of hologram spectrum in Figure 11 restored from a limited area; (**d**) second step of hologram spectrum in Figure 11 restored from a limited area; and (**e**) second step of hologram spectrum in Figure 11 restored from a limited area.

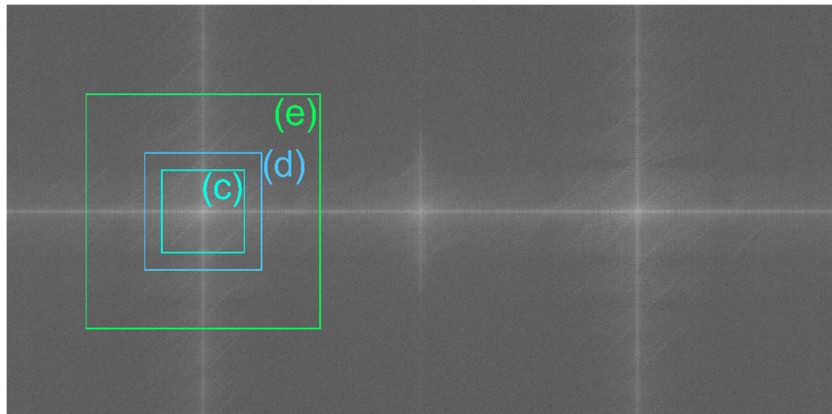

**Figure 11.** Spectrum of hologram synthesized from object in Figure 9: selected areas equal to number of pixels (**c**), one and a half times the number of pixels (**d**), and eight times the number of pixels of the holographic object (**e**).

It can be seen that the first algorithm (Figure 10c–e) with the transmission of the hologram spectrum over the communication channel, equal to and even larger in size than the area of the texture + mask frame pair, is inferior to the second algorithm (Figure 10b) with the transmission of resolution of the texture + mask pair over the communication channel. Moreover, in the first case, an increase in the data volume by almost an order of magnitude (eight times) when transmitting the spatial spectrum of the hologram (Figure 11e) still does not give the same resolution of the restored image as eight times smaller (Figure 11) in terms of the information transfer algorithm for texture + mask with subsequent syn-

thesis of the hologram already at the receiving end of the communication channel and reconstruction of the 3D image of the object from this hologram (Figure 10b).

### 2.5. Holographic Information Transmission Rate

In addition to exploring the above-mentioned fundamental possibility of transmitting a 3D holographic frame by transmitting two 2D frames, a simple experiment was carried out to transmit their sequence over a wireless Wi-Fi communication channel to simulate 3D video. To do this, the FTP (file transfer protocol) protocol was installed on the transmitting device for transmitting frames, and the FileZilla program for working with FTP servers was installed on the receiving device. Based on [37], each transmitted frame of a 3D image was the sum of two 2D frames, a texture (2000 × 2000 pixels) and a mask (1000 × 1000 pixels). To simulate the transmission of a video sequence, packages of 500 double frames (texture + mask) were simultaneously transmitted. The transmission time of these frame packages measured by the FileZilla program during real-time playback showed that the transmission of complete holographic information of a 3D object in real time had a frame rate of more than 25 frames/s, which is quite feasible [45].

Table 1 shows the options for presenting these pairs of images.

**Table 1.** Variants of representation of pairs of 2D frames.

| Image Format | File Size with Texture (2000 × 2000 Pixels) (kB) | File Size with Mask (1000 × 1000 Pixels) (kB) | Total Volume of 3D-Frame (kB) |
|---|---|---|---|
| BMP | 3908 | 978 | 4886 |
| PNG | 593 | 67 | 660 |
| JPEG (70% quality degree) | 127 | 25 | 152 |

Table 2 shows the results of direct measurement of the transmission of pairs of such frames over a Wi-Fi channel.

**Table 2.** The results of measurements of the transmission time of pairs of frames over the Wi-Fi channel.

| Parameter | Symbol | File Format with Frame | | |
|---|---|---|---|---|
| | | JPEG | PNG | BMP |
| Number of transmitted frames, texture + mask | $N$ | 500 frames | 500 frames | 500 frames |
| Total amount of data (MB) | $Vol$ | 74.11 | 322.43 | 2385.46 |
| Transfer time (s) | $t$ | 12 | 51 | 390 |
| Transfer rate (MB/s) | $V = \frac{Vol}{t}$ | 6.18 | 6.32 | 6.12 |
| Frame rate during playback in real time (frames/s) | $\nu = \frac{N}{t}$ | 41.67 | 9.80 | 1.28 |

The results show that the transmission of texture + mask frame pairs is quite feasible by conventional means of communication, and this method can be used in tasks of holographic TV and augmented reality.

It can be seen that the JPEG format, which is three-step signal processing, is better suited for the purpose of transmitting video information. The first step is line scanning (three colors) of the image and its discrete representation. The second is discrete calculation of the spectrum and the limitation of its high-frequency components. The third is encoding the series lengths by methods that increase entropy, such as the Huffman method. The easily obtained frequency of more than 40 frames per second is not the limit, since the encoding of the texture and mask occurs separately. The Huffman method, as an additional compression, can also be successfully applied to the pairs of texture + mask images considered in this paper, each of which can have its own size and sampling, so it may well give even better compression for a higher transmission rate.

## 2.6. Hyperspectral Images and 3D Image Multiplexing

Figure 12a,b show synthesized 2D and 3D images of a human portrait formed digitally from three restored gray holograms, each of which transmitted 3D data via one of the three RGB (red, green, blue) channels at wavelengths of 0.633, 0.532, and 0.435 μm. Figure 12c shows a 3D object image restored at the receiving end of a telecommunication channel in the visible range, which was recorded at the transmitting end of a telecommunication channel in the IR range of electromagnetic radiation at a wavelength of 0.937 μm, which corresponds to the radiation temperature of a human body. It is apparent that the suggested technology for transmitting holographic information enables us to combine color channels not only within the visible range but also far beyond it, and by doing so, the captured spectrum bandwidth can be enhanced. Moreover, this is done economically, by transmitting a frame of an image mask and its color coding frames. Therefore, it takes only four frames to transmit an RGB 3D image, and only five frames to transmit a hyperspectral image that includes an IR line.

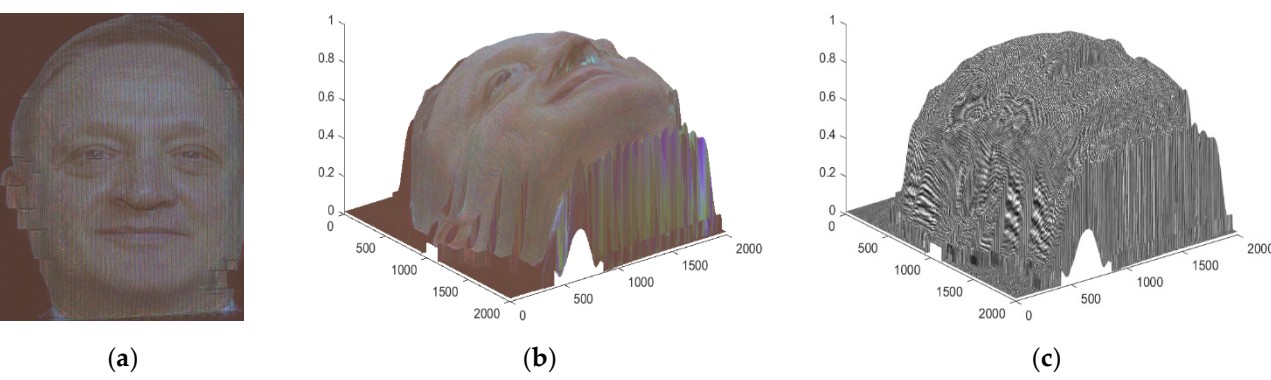

(**a**)  (**b**)  (**c**)

**Figure 12.** (**a**) 3D colored 2D image, restored under RGB standard at receiving end of telecommunication channel; (**b**) 3D colored 3D image, restored under RGB standard at receiving end of telecommunication channel; and (**c**) half-tone (gray) IR image restored in visible range at receiving end of telecommunication channel.

Since the diffraction structure of the bands, called a hologram, is already formed at the receiving end of the communication channel, the described method of transmitting holographic information of a 3D object is well suited for restoring several three-dimensional images at different depths of the image space at once without changing their transverse scale. This is due to the representation of wavelength λ and the depth of space Z in the kernel of the Fresnel transformation Equation (10) as a product ($k/z = 2\pi/\lambda z$) and the ability to digitally correct a hologram recorded at one wavelength and restored at another. At the same time, holograms synthesized using a single mask but with several textures in different colors on one hologram will restore the same number of monochrome images of different colors separated in space. The digital correction of the hologram can be placed on the same axis (Figure 13).

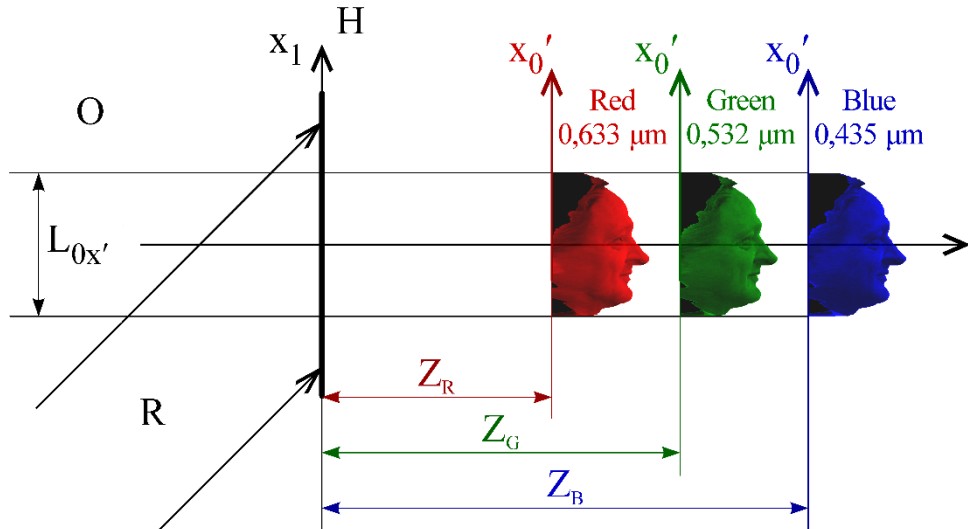

**Figure 13.** Restoration of three monochrome images by one hologram.

The opposite is also true: the same 3D image can be reconstructed from three monochrome holograms from different distances (Figure 14), providing the possibility of forming moving color 3D images, including in the space of real images in front of the red monitor screen.

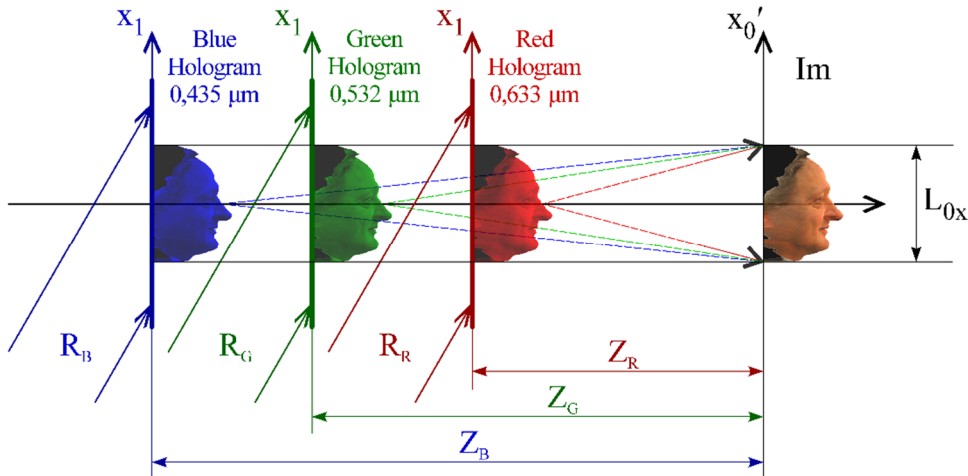

**Figure 14.** Synthesis of color image of 3D object by three dynamic holograms.

By expanding this method, it is possible to organize the restoration of multicolor images recorded at wavelengths wider than the visible range. When restoring such holograms with the appropriate wavelengths, hyperspectral 3D images are formed (Figure 15). Additionally, in the proposed technology, it is relatively simple to implement the shift of all restoring hologram waves in order to compress the spectrum of the restored image, placing a wider range in the visible range; for example, moving the IR image to the red part of the spectrum, which, accordingly, can be shifted closer to the green.

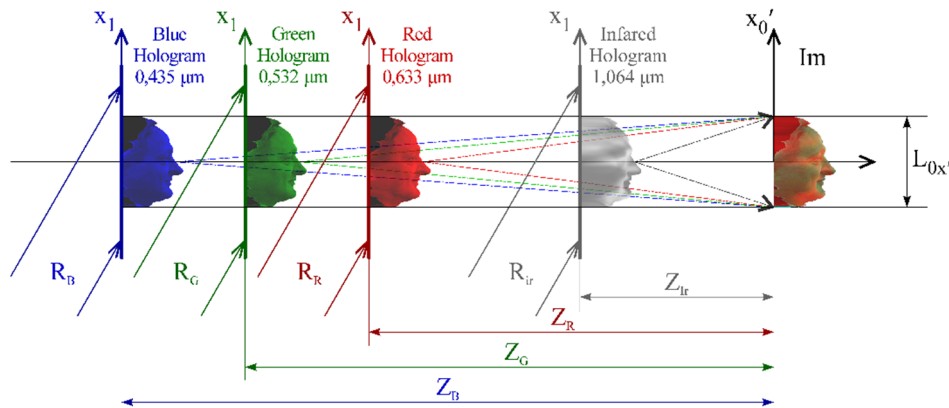

**Figure 15.** Synthesis of hyperspectral 3D images.

An example of such virtual movement of a restored 3D image along the optical axis is shown in Figure 16.

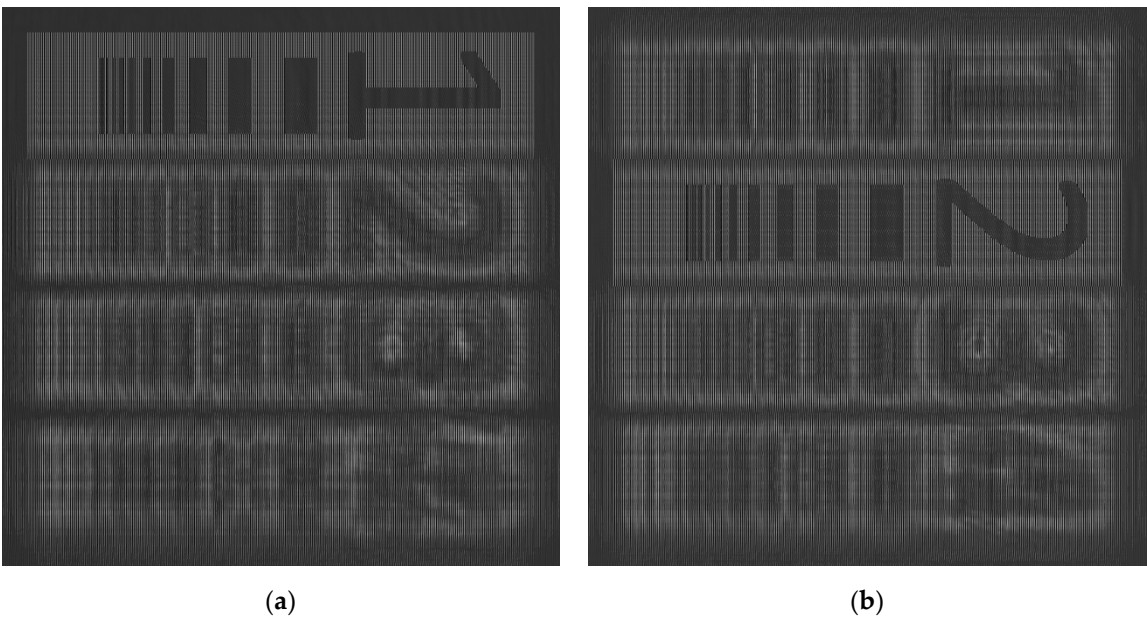

(**a**)                                    (**b**)

**Figure 16.** (**a**) Object reconstructed at λ = 0.532 μm at a distance of Z = 59 mm; (**b**) object reconstructed at λ = 0.633 μm at a distance of Z = 70 mm from the hologram.

It can be seen that sharp reconstruction distances increase in proportion to the length of the reconstruction wave, 59 × (0.633/0.532) = 70.2 mm, which in the presented case corresponds to the position of the second step.

When forming multispectral images, multiplexing is easy to implement, since it is enough to transmit one surface mask for different textures. Thus, the described technology of creating a hologram of a 3D object at the receiving end of the communication channel is consistent with the tasks of aggregating and forming multispectral images, although each of these tasks is separate and complicated work.

### 3. Experiment

Full-scale experiments on synthesis of material, dynamically changing holograms are currently impossible due to absence of dynamic holographic displays. Based on the above-described computations, however, a material hologram was recorded in DotMatrix technology, using the KineMax MASTERING SYSTEM (Polish Holographic Systems, Warsaw, Poland). The hologram was synthesized at the receiving end of the communications

channel and a 3D image of a human being was restored in the space between the holographic screen and the observer (Figure 17). It is visible on the shifting of two material needles: one is located farther and "touches" an ear of the restored image, and the other is located close to an eye. Changing the observation angle of the 3D object, the distance between the needles changes, although the tip of each needle aligns with a point of the object "hanging in the air" with the accuracy to aberrations. For instance, the parallax and the outreach of the 3D image in the observer's space are seen.

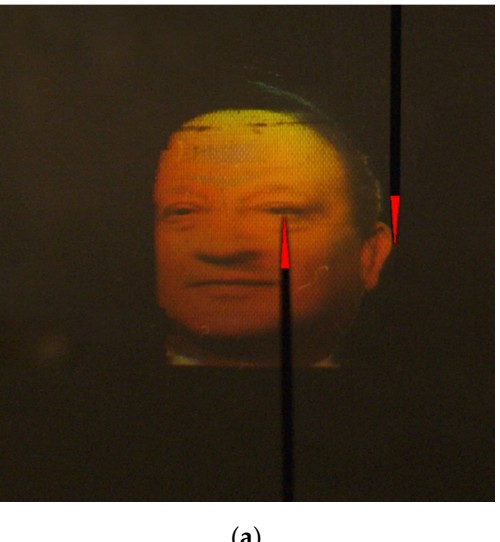 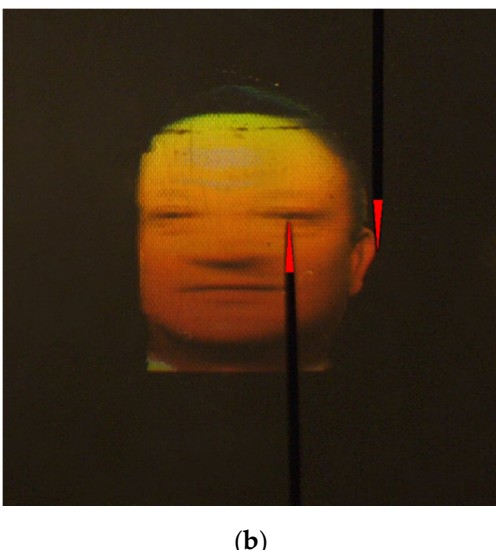

(**a**)            (**b**)

**Figure 17.** The 3D human portrait image restored from an experimentally created hologram based on Figure 6: (**a**) right profile, (**b**) full face.

Therefore, in view of the material from Section 2.5, it can be taken that possibility of transmitting 3D holographic information via a conventional communications channel is demonstrated experimentally. So far, the speed of hologram synthesis lags behind the transmission speed. However, it is a computational problem (unlike transmission problems) and it can be solved, for example, by parallel computations. We are working on the requirements for the necessary equipment. Regardless of that, possibility of transmitting 3D dynamic holographic information via a conventional communication channel and restoring such a 3D image in the observer's space, in front of a holographic monitor, can be considered proven.

### 4. Main Conclusions

This paper shows the possibility of implementing a full cycle starting with the recording of a 3D image of a living object in real time by registering two 2D images (mask + texture), and then transmitting them over a conventional Wi-Fi channel. This method is, in essence, physically close to the SSB method, but the 3D image restored at the receiving end of the communication channel has higher resolution. This shows the possibility of practical implementation of holographic television and augmented 3D reality systems and bringing to life holographic 3D TV and augmented 3D reality if a dynamic holographic display is available.

The method used in this paper for recording 3D information of a living object, and restoring a 3D image of the holographic object at the receiving end of the communication channel not only allows us to solve the problems of holographic cinema, television, and augmented reality but also can be used in tasks of 3D image telecomplexing and multispectral 3D image teleforming.

The properties of the Fresnel transformation, where wavelength λ is in the product with longitudinal distance Z, where the 3D image is restored, allows us, on the one hand,

to restore equal scale (x, y) 3D images at different distances Z from the hologram and, on the other hand, to collect such images from different holograms into one 3D image, additionally spending on its transmission a small part of the radio signal spectrum equal or close to the spectrum of one 2D image. Such cost-effective complexing of multispectral images can be of interest in a number of medical, biological, and technical fields [46].

Amplitude and phase modulation of the carrier spatial frequency manifest in holography in a particular way, as the harmonics influence the signal structure not directly but through a Fresnel transform. Forming spatial frequencies in holography is similar to forming time frequencies used in the SSB method, but differentiated by a more complex Fresnel transform compared to the Fourier transform, which is simply convenient to use in the descriptive and computational part.

The possibility of transmitting such a 3D signal with an image of a portrait of a person with a generally accepted TV frame rate was proved with the help of direct simulation, which was about measuring the time required for transmitting 500 pairs of frames (texture + mask).

The nature of carrier frequency formation and deviation are different. The carrier frequency is defined by a convergence angle of the object and reference beams in such a way that, at a certain distance, the restored object image in the minus first order diffraction beam can spatially diverge with the zeroth order of the restoring beam that passed through the hologram. Spatial frequencies that cause deviation in the carrier frequency are formed in the object wave and have another reason, as they are generated with Fresnel diffraction on the object. The nature of these spatial frequencies differs, and the difference between them is significant, which can be used (and was used by us in this work) when compressing holographic information.

Expanding the scope of the proposed method, it is possible to set the task of recording a 3D image in acoustics, and restoring it in the visible or other range of the electromagnetic spectrum, and vice versa. The first is an urgent task, such as in medicine, and the second in various technical fields, such as in the processing of materials.

### 5. Conclusions

In this paper, the possibility of a full cycle (i.e., from recording a 3D image of a human with Full HD quality and transmitting this information via conventional communications channel to its restoring into a holographic 3D image) is proven theoretically and experimentally. The authors do hope that the solution for the problem of transmitting 3D holographic information via conventional communication channels presented in the paper will add an incentive to solve the problem of dynamic holographic monitor synthesis as the last problem on the way to holographic TV and 3D augmented reality (Video S1).

The authors express their gratitude to the Research Institute of Radio Electronics and Laser Technology of Moscow State Technical University and personally to I. Tsyganov and E. Drozdova for the experimental hologram synthesis for the purposes of this study.

**Supplementary Materials:** The following are available online at https://www.mdpi.com/article/10.3390/photonics8100448/s1. Video S1: Aggregation from different carrier frequencies.

**Author Contributions:** Conceptualization, S.A.S.; methodology, S.A.S.; software, A.L.P.; validation, A.L.P.; investigation, S.A.S. and A.L.P.; resources, S.A.S.; data curation, S.A.S.; writing—original draft preparation, S.A.S.; writing—review and editing, A.L.P.; visualization, A.L.P.; supervision, A.L.P.; project administration, S.A.S. All authors have read and agreed to the published version of the manuscript.

**Funding:** This research received no external funding.

**Institutional Review Board Statement:** Ethical review and approval were waived for this study, due to REASON (The texture, mask and 3D images used are personal images of the first author of this article, Sergey Alexandrovich Shoydin. Sergey Alexandrovich Shoydin hereby declares that his images can be used in this article. Authors of other scientific articles can also use the 3D images, mask

and texture with the image of Sergey Alexandrovich Shoydin used in this work in their scientific articles with the obligatory reference to this work).

**Informed Consent Statement:** Informed consent was obtained from all subjects involved in the study.

**Data Availability Statement:** Data are available from the authors upon request.

**Acknowledgments:** The authors thank all reviewers for their helpful comments and suggestions.

**Conflicts of Interest:** The authors declare no conflict of interest.

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
