# Peer review of "Transmission of 3D Holographic Information via Conventional Communication Channels and the Possibility of Multiplexing in the Implementation of 3D Hyperspectral Images"

_photonics, doi:10.3390/photonics8100448_

Round 1

Reviewer 1 Report

This manuscript suggests the method to transmit 3D holographic information via a conventional WiFi channel in real-time. The motivation and the idea itself are convincing, and the related works are worth to be announced to the public. The manuscript presents various descriptions to process and transmit the hologram data, and the author seems to devote hard work to writing the manuscript as far as possible.

Even though the author's huge effort on this manuscript, I am sorry to inform you that the manuscript is too lengthy to clearly read out the primary contribution of the study. And, although there are plenty of descriptions to process the hologram information, most of the theories and ideas seem to be found in the textbook and many other studies which might be redundant in technical papers. Speaking about the figures and supplementary video, I think many of them are not really necessary to deliver the message of the manuscript, rather they make me confused to follow the flow of the story. Therefore, I have no choice but to ask the author to refine and reorganize the paper structure to deliver a clear message to the readers.  

And, as the author mentioned, the work on this manuscript only suggests the issues and some basic numerical simulation about the holographic data broadcasting, which seems not enough to be published as a scientific paper. Therefore, I request the author to conduct a proof-of-concept experiment based on the proposed methods, then re-submit the paper. 

Author Response

The authors are grateful to all reviewers for their valuable comments.

Reviewer 2 Report

This paper is about a new method to transmit 3D holographic information, by transmitting two 2d images including depth map and surface texture. I think this is a key technology for future holographic telecommunication, and the authors show their expertise in this area by presenting many details, experiments, and a very long manuscript. I saw this manuscript provides enough information about this topic, but there are too much redundant information. Moreover, there are many nonstandard expressions, e.g., lack of full expressions after an abbreviation and the structure of this manuscript seems different from an academic paper. This work can be published only if the authors rewrite and reorganize it.

Author Response

(The authors gave the same response as above.)

Reviewer 3 Report

This paper describes a method for holographic bandwidth compression over network transmission via the transmission of other primaries to the image content (RGB texture + depth map) that are already highly amenable to conventional compression schemes, and in contrast to a fully-computed Fresnel CGH, are much less redundant in information content and efficient for transmission.

Overall, I think this paper is well-written and the methods, motivations, and results are well-described. I have a few comments/questions for which the authors may include additional text:

1) The synthesis of the hologram at the receiving end from the RGB + depth images is described in detail in Section 2.2; however, I didn't see much in the way of other (perhaps, more economical and less compute-intensive) algorithms besides the Fresnel transform approach. Has this approach been tested for real-time scenes, and if so, what is the performance in terms of latency, frames per second, etc.? I wonder about the comparison in those metrics, and also overall image fidelity, relative to approaches that use holographic stereogram techniques (e.g., the authors could consider the approach in

J Barabas et. al. "Diffraction specific coherent panoramagrams of real scenes", Proc. SPIE 7957, Practical Holography XXV: Materials and Applications, 795702 (3 February 2011); https://doi.org/10.1117/12.873865) 

as a point of reference for alternative methods.

2) I am also curious about the applicability of phase unwrapping techniques to the layer-based Fresnel methods described here. What is the added benefit of using phase unwrapping as described in the last paragraph of Section 2.3?

Author Response

(The authors gave the same response as above.)

Round 2

Reviewer 1 Report

I deeply apologize that my judgment on the previous review process may have offended the author. The authors put a lot of effort into improving the quality of the submitted manuscript after then, and kindly answered to the reviewer's questions. 

However, while reading the revised manuscript, I still cannot get rid of the thought, unlike other lengthy and heavy papers, it is verbos and there are lots of redundant explanations. Another reviewer have also pointed out this problem, so I don't think my opinion is completely wrong. 

Nevertheless, I hope that this paper will be open to the public, referring to the value of the research conducted by the author and his efforts to inform it. 

One request: Please decribe the method to fabricate the analog hologram in experimental section in short.  

Reviewer 2 Report

I didn't see enough improvements. The work is good but the paper is not in a good level. The authors may re-organize it and re-submit.

This manuscript is a resubmission of an earlier submission. The following is a list of the peer review reports and author responses from that submission.

Round 1

Reviewer 1 Report

After revision, the resubmitted manuscript has been significantly improved. It can be accepted for publication. Some minor revision suggestions: for the relaionship between JPEG and hologram compression, the following work [r1] shall be included in the literature review; for fast hologram calculation from a point cloud, the follwoing work [r2] shall be included in the literature review. [r1]S. Jiao, Z. Jin, C. Chang, C. Zhou, W. Zou, and X. Li, “Compression of phase-only holograms with JPEG standard and deep learning,” Applied Sciences, 8, pp. 1258. (2018) [r2]D. Pi, J. Liu, Y. Han, A. U. R. Khalid, and S. Yu, "Simple and effective calculation method for computer-generated hologram based on non-uniform sampling using look-up-table," Opt. Express 27(26), 37337-37348 (2019)

Reviewer 2 Report

None of the concerns mentionned in my previous review was removed by the provided updates. 

We recognize that significant efforts have been made by the authors to provide more details on specific sections and answer reviewers remarks. However, the explainations given as well as the updates in the articles confirm that the proposed method does not concern hologram compression but rather Depth+Texture compression, for which dedicated methodology (and related work review) should be considered.

Reviewer 3 Report

The technical proposal seems reasonable and the science is sound.

The paper can be accepted for publication, however, there a serious issue with the English quality.  The paper is made difficult to read.  The English  while not terrible is "clunky" and very difficult to follow in places.

I note in several equations seem to have unintended symbols, e.g. (9) (10 and (111).  This may be an issue with my software.

I also think the paper could have been a bit more concise.  In particular the abstract seems too long.  A shorter statement of what is in the paper would help the reader.